# SPR-Based Kinetic Analysis of the Early Stages of Infection in Cells Infected with Human Coronavirus and Treated with Hydroxychloroquine

**DOI:** 10.3390/bios11080251

**Published:** 2021-07-26

**Authors:** Petia Genova-Kalou, Georgi Dyankov, Radoslav Marinov, Vihar Mankov, Evdokiya Belina, Hristo Kisov, Velichka Strijkova-Kenderova, Todor Kantardjiev

**Affiliations:** 1National Center of Infectious and Parasitic Diseases, 44A “Gen. Stoletov” Blvd., 1233 Sofia, Bulgaria; petia.d.genova@abv.bg (P.G.-K.); r.m.r@mail.bg (R.M.); nrlcell_rik_herp@ncipd.org (T.K.); 2Institute of Optical Materials and Technologies “Acad. J. Malinowski” (IOMT), Bulgarian Academy of Sciences (BAS), 109 “Acad. G. Bonchev” Str., 1113 Sofia, Bulgaria; viharmankov@gmail.com (V.M.); evdokiyabelina@yahoo.de (E.B.); christokissov@yahoo.com (H.K.); vily@iomt.bas.bg (V.S.-K.)

**Keywords:** SPR, cell-based assay, viral growth kinetics, human coronavirus, hydroxychloroquine

## Abstract

Cell-based assays are a valuable tool for examination of virus–host cell interactions and drug discovery processes, allowing for a more physiological setting compared to biochemical assays. Despite the fact that cell-based SPR assays are label-free and thus provide all the associated benefits, they have never been used to study viral growth kinetics and to predict drug antiviral response in cells. In this study, we prove the concept that the cell-based SPR assay can be applied in the kinetic analysis of the early stages of viral infection of cells and the antiviral drug activity in the infected cells. For this purpose, cells immobilized on the SPR slides were infected with human coronavirus HCov-229E and treated with hydroxychloroquine. The SPR response was measured at different time intervals within the early stages of infection. Methyl Thiazolyl Tetrazolium (MTT) assay was used to provide the reference data. We found that the results of the SPR and MTT assays were consistent, and SPR is a reliable tool in investigating virus–host cell interaction and the mechanism of action of viral inhibitors. SPR assay was more sensitive and accurate in the first hours of infection within the first replication cycle, whereas the MTT assay was not so effective. After the second replication cycle, noise was generated by the destruction of the cell layer and by the remnants of dead cells, and masks useful SPR signals.

## 1. Introduction

The cell is the minimum functional unit of living organisms. Knowledge of the basic cellular components and the way cells work is fundamental to life sciences, including molecular biology, cell biology, cell physiology, etc. With the traditionally used cell-based assays, it is a common practice to use radioactivity, chemiluminescence, or fluorescence to produce a measurable signal. Label-free cell-based assays have sparked interest due to their ability to measure cell responses without additional reporter compounds. Among the different label-free techniques, optical methods have been widely adopted for cell-based assays. The most effective one—surface plasmon resonance (SPR)—has been applied in the study of a variety of cellular processes.

In its conventional approach, SPR detects the binding of molecules in the detection volume on a sensor chip in real-time without any labeling. The signals are generated by a change in the biomolecule layers and are linearly related to their thickness. This is true in the first approximation since the layers are uniform and much smaller than the penetration depth of the plasmon wave. The situation is different in cell-based SPR assays where cells are immobilized on the sensing surface and serve as sensing elements. Nevertheless, SPR sensing has been extended into a powerful method for sensing large biological objects such as cells [1].

### 1.1. Penetration Depth and Detection Depth

An essential concept in SPR sensing is the nature of the evanescent field of the plasmon wave. This is especially relevant in functional cell-based assays because the penetration depth of 150−500 nm from the metal surface is only a fraction of the height (vertical dimension) of commonly used cells, which is in the range of several micrometers. Thus, it was suggested that the SPR signal is provoked by biological events in the area near the plasma membrane, whereas events inside cells, especially in their upper area, cannot be detected [2,3,4].

Two approaches have been used to achieve a greater penetration depth of the plasmon wave: excitation of long-range surface plasmon in a specific SPR biochip [5] and plasmon excitation in the UV range [6,7]. The latter approach seems to be more effective—the penetration depth could reach several microns. Even though these modifications are clearly advantageous, it may turn out that the more distant cell regions can be detected without applying them.

Although in the majority of cases the penetration depth is within the range of several hundred nanometers, SPR sensing is not necessarily applied to this limited range. SPR has been successfully used in detecting cell responses to external triggers such as drugs [8,9,10,11] and external stimuli [12].

It has been demonstrated [1,13,14] that a refractive index (RI) near the plasma membrane might reflect the accumulation and rearrangement of proteins activated by intracellular signal transduction provoked by exogenous stimuli. The SPR signals generated by the cellular response originate from complex biological events that have a local impact on RI. Additional experiments are required to find out what biological matter elicits the SPR signal.

### 1.2. Application in Drug Research

SPR technology has been widely applied in studying drugs. These studies have been generally limited to bimolecular binding assays outside living cells where purified biomolecules have been immobilized and a binding reaction with the target drug molecules has been detected [15,16,17].

Instead of immobilizing purified biomolecules in binding assays, a whole-cell adhesion to a sensing surface would provide on-site signals from drug–living cell interactions. Therefore, the pharmacokinetic parameters obtained by the cell-based assays would be more accurate and reliable than those obtained by the biochemical binding assays. A number of research groups have reported cell-based SPR assays used in evaluating the efficiency of a variety of drugs. A comprehensive review of cell-based SPR assay can be found elsewhere [18].

### 1.3. Cell-Based SPR Assay in Virus Research

The SPR technique has been widely used in studying viruses. Its well-known advantages are as follows: (i) the fact that it is label-free, thus eliminating functionalization of multiple antibodies, which occurs with ELISA; (ii) dynamic measurement of binding–unbinding kinetics; and (iii) high sensitivity, providing reliable virus detection. Reasonably, more research has been focused on viral diagnosis. The recognition elements used have included antibodies, antigens, DNA, and aptamers. Viral kinetic analysis of cells infected with SARS [19], SARS-CoV [20], and SARS-CoV-2 [21] has been performed. Surprisingly, the cell-based SPR assay has never been used in virus research so far. 

### 1.4. Aim of the Present Study

Coronaviruses are disease-causing agents that infect many species of mammals and birds. Some, such as HKU1, OC43, 229E, and NL63, circulate seasonally and cause respiratory diseases in children and adults, which are not life-threatening. At the end of 2019, the identification of SARS-CoV-2 as the causal agent of atypical pulmonary diseases was the latest example of these emerging coronaviruses. It is essential to investigate the way in which the virus replication cycle occurs. MTT and immunofluorescence have been widely applied in investigating virus kinetics. However, the necessity of studying virus kinetics in the first hours of infection requires the application of new methods. Although the cell-based SPR assay has been successfully used to study intracellular processes, it has never been applied in examining the kinetics of the ultrastructure of virus-infected cells. The aim of this study was to prove that the cell-based SPR assay can be applied in the kinetic analysis of the early stages of viral infection of cells and the antiviral drug activity in the infected cells.

## 2. Materials and Methods

There are seven human coronaviruses (HCoV), highly diverse and causing respiratory diseases with mild to severe outcomes [22]. Currently, no specific antiviral drugs to treat HCoV infection are available, although hydroxychloroquine (HCQ) has been suggested as appropriate [23]. The way in which HCQ exerts its effect on time-dependent HCoV growth is well known and helps us analyze the SPR signal and compare it with other methods.

*Human cell line culture:* HCoV, strain 229E (HCoV-229E) was isolated using Vero E6 (African green monkey kidney) cell line, obtained from the National Center of Infectious and Parasitic Diseases (NCIPD), Bulgaria. The cells were cultured in Dulbecco’s Modified Eagle’s Medium (DMEM, Sigma-Aldrich, Sent Luis, MO, USA), supplemented with 10% fetal bovine serum (FBS, Gibco^TM^ by Life Technology, Darmstadt, Germany) and antibiotics at 37 °C, 5% CO_2_ atmosphere.

*Virus propagation:* The HCoV-229E (from the NCIPD viral collection) was propagated in Vero E6 cells that reached 70% confluence in DMEM media, supplemented with 2% FBS at 37 °C, 5% CO_2_ atmosphere.

*Virus titration:* Confluent Vero cells (3 × 10^4^ cells/well) were cultured in 96-well plates (100 μL/well). Serial 10-fold dilutions of the HCoV-229E stock (10^−1^ to 10^−8^) were prepared in DMEM supplemented with 2% FBS, at 37 °C and 5% CO_2_ atmosphere for 4 days. Cell viability and yields of virus progeny were measured post-infection (*p.i.*) every 24 h for a 96 h period of total incubation time. The infected cells were monitored microscopically daily for cytopathic effects (CPE) in the infected cells caused by HCoV-229. The titer of the purified HCoV-229E was 10^4.5^ (high-titer) 50% tissue culture infection doses (TCID(50)/mL.

*MTT assay:* The MTT assay (3-(4,5-dimethyl-2-thiazolyl)-2,5-diphenyl-2H-tetrazolium bromide (Methyl Thiazolyl Tetrazolium; MTT) is used to measure cellular metabolic activity as an indicator of cell viability, proliferation, and cytotoxicity. MTT is reduced by mitochondrial dehydrogenases to the water-insoluble pink compound formazan, depending on the viability of cells. Vero cells were seeded in 96-well microtiter plates (3 × 104 cells/mL) and infected with HCoV-229E, multiplicity of infection (MOI) 0.1, and treated with different HCQ non-toxic concentrations at different hours. Measuring the optical density (OD) by the MTT assay has been used as a sensitive method to quantify the density of the HCQ-treated infected cells [24]. The OD values have been measured at a wavelength of 540 nm using ELISA reader (Sunrise Basic Tecan, Männedorf, Switzerland), whereby the concentration of viable cells is found [25]. The same approach was used in our study as well.

*Cell-based SPR assay:* The SPR slides were derived from a recordable compact disc (CD-R). A gold layer with thickness 80–100 nm was deposited onto the polycarbonate substrate by vacuum thermal evaporation. 

Before seeding the cells, the slides were immersed in isopropyl alcohol and cleaned ultrasonically for 10 min. Then, they were rinsed thoroughly with high purity water, dried, and illuminated by UV light for 24 h.

The Vero E6 cells were cultured in DMEM (Dulbecco’s Modified Eagle Medium), supplemented with 10% heat-inactivated fetal bovine serum (FBS) and antibiotics at a density 3 × 10^4^ cells/mL and incubated for 24 h at 37 °C and 5% CO_2_ conditions to allow cell adhesion to the SPR slide surface. When the cells achieved appropriate density (about 70% confluence on the SPR slide surface), a monolayer was washed twice with phosphate buffer solution (PBS), pH = 7.3, the supernatant was carefully removed, and the cell culture medium was supplemented with 2% FBS and HCoV-229E with multiplicity of infection (MOI) 0.1. After virus adsorption, the infected cells were treated with a non-cytotoxic concentration (1 mg/mL) of the antiviral drug HCQ. The infected and treated cultures were incubated at 33 °C in a humidified 5% CO_2_ atmosphere. Cell morphology was observed every 6 h for microscopically detectable morphological alterations, such as loss of confluency, cell rounding and shrinking, and cytoplasm granulation and vacuolization. The viability of the infected and treated cells from each well of the 96-well culture plate was determined every 6 h by an MTT-assay, and the SPR spectral shift of the cell-based SPR assay was also measured.

*SPR measurement:* The gilded diffraction grating is part of a continuous CD-R spiral groove with a period of 1.55 μm. A Θ-2θ goniometer with a 0.01 deg accuracy was used for the SPR excitation and registration. Spectral interrogation was used for the SPR registration. A collimated beam of p-polarized white light under angle of incidence in the range 35–42 degrees excited resonances between 710 nm and 610 nm in a bare grating. A spectrometer registered the spectrum in the zero-order reflection. The optical setup is depicted in Figure 1.

Figure 2 shows the experimentally observed resonances in the bare SPR slides and the cell-based SPR assays: a slide with cells obtained at 12 h after seeding and a slide with infected cells at 12 h *p.i.* The changes in cellular morphology, which in turn led to a variation of the effective refractive index at the interface between the cell membrane and the metal layer, caused a well observable spectral shift, marked in Figure 2. Reference resonances were established for bare grating—the black curve in Figure 2. The spectral shift of the slides with non-infected cells was evaluated against reference resonances—marked as “A” in Figure 2. The spectral shift established in this way is referred to as “cell control”. The SPR shift for infected cells was evaluated against the cell control—marked as “B” in Figure 2. The spectral shift established in this way is referred to as “virus control”. The SPR spectral shift of the treated cells was evaluated against the virus control, after which it was compensated for by dividing the difference between cell control and virus control. The signal established was referred to as “SPR compensated signal”. The SPR responses were measured at different time intervals between 4 and 48 h.

*AFM examination:* An Atomic Force Microscope (AFM) (Asylum Research MFP-3D (Oxford Instruments) was supplied with silicon nitride probes: frequency of 30 kHz, spring constant of 0.27 N/m, and radius <15 nm. The experiments were carried out at ambient conditions using the AFM contact mode. For the purposes of scanning, the cell-based SPR assays were fixed with 2.5% glutaraldehyde.

In the next section, we describe what type of biological events occur at different moments and determine the registered SPR shift.

## 3. Results

### 3.1. Cells Growth Kinetics

Non-human primate kidney cell line Vero E6 was seeded at concentration 3 × 10^4^ cells/mL on SPR slides and on glass plates. The cells on the glass plates were counted by MTT. Every 6 h, the SPR spectral shift of the cell-based SPR assay was measured. The results are presented as a dotted line in Figure 3. A reference measurement with an MTT assay was provided. The data obtained—the mean values of three independent experiments—are presented in Figure 3: an SPR assay measurement (dotted line) and an MTT measurement (solid line). Obviously, the SPR signal follows the temporal change of cell viability established by the MTT assay. We also observed that the Vero E6 cells grew rapidly, producing a confluent monolayer. Even after a prolonged period, the Vero E6 cells showed the typical morphological characteristics of spindle-shaped fibroblasts—flat, without prominence in their shape and surface, with intact cytoplasm and oval nucleus, 15–20 μm long, and about 5 μm high—all of which was microscopically proven by the AFM examination (Figure 4). The AFM study showed that the cells had adhered tightly to the grating surface, which ensured an effective penetration of the plasmon wave into the cells. The population doubling per day (Pd/D) and cell density were found to increase with the prolongation of cell cultivation expressed as a steeper SPR curve in the range 25–38 h (Figure 3). The growth curves constructed from both the SPR and MTT assays showed the typical pattern of a growth curve. The SPR curve clearly indicated a lag phase (until 20 h), an exponential phase (25–38 h), a plateau (around 40 h), and reaching a death phase (40–48 h). This suggested that the cell density used was appropriate for running the experiments with the present virus.

### 3.2. Viral Growth Kinetics

To estimate the viral growth curve characteristics, the infectivity titer of HCOV-229E was determined every 6 h *p.i.* by MTT (solid curve in Figure 5). The MTT assay measurement revealed the main phases of the viral growth kinetics. The initial increase in viability by 10 h is due to an increase in the number of uninfected cells as a result of the cell growth process.

The increase in the SPR spectral shift (dotted curve in Figure 3) lasted until 18 h, probably due not only to the increased cell density on the grating surface but also to the attachment of the viruses to the cell membrane. Then, the SPR assay measurement accurately indicates the infection efficiency.

We would like to point out that the measurement time interval of MTT assay was almost twice as long as the SPR time interval. This is due to the lower time resolution of MTT. The highest time resolution of SPR assay can explain observed local maximums of the cell control around 12 h (Figure 3) and of virus control around 18 h (Figure 4).

MTT showed a substantial decrease in cell viability in the interval 20–24 h (Figure 5), which corresponded to dramatic ultrastructural changes—marked granulation of cell cytoplasm, particularly around the nucleus with the fragmentation of the latter, and proliferation of pseudopodia at the cell periphery. This was confirmed by the AFM examination (Figure 6).

The SPR signal also decreased in this interval and reached its minimum around 30 h. This is close to the minimum of cell viability (MTT curve—Figure 5), well pronounced in the interval 30–40 h, confirming the cell monolayer destruction, cell deterioration, and detachment from the surface as a result of the increased virus production.

At the end of the first replication cycle (24–30 h), the virus was expressed in the intercellular space. This was clearly observed by the AFM study as shown in Figure 7, which represents a magnified part of an area (shown in Figure 4) located near the cell membrane. Viruses (marked with arrows) have just been expressed from the host cell at 24 h p.i and are still close to the cell membrane. After this moment, the second replication cycle starts: virus attachment to the cell membranes of new cells. As a result of this, compaction of the cell membrane occurred, and the refractive index increased. This coincided with the increase in the SPR signal after 30 h.

The change in the MTT signal was more inert—it increased after 36 h due to the competing processes of cell growth and viral replication.

### 3.3. Kinetics of Antiviral Activity of HCQ

Hydroxychloroquine (HCQ), used to treat malaria and some autoimmune disorders, might be of certain use in the clinical management of infections caused by HCoV, especially SARS coronavirus (SARS-CoV-1) and SARS-CoV-2, by potently inhibiting the infection, a fact that has been found in cell culture studies [26]. Here, we report an in vitro kinetic analysis of the antiviral activity of HCQ against the HCoV-229E strain. We would like to point out here that such a study has not been carried out so far.

The cytotoxicity of HCQ in Vero E6 cells was measured in advance for the purposes of determining the concentrations that would not cause injury or death to the treated cells (data not shown). This experiment was conducted three times and the results obtained are shown in Figure 8. To gain an initial insight into the stages of the viral replication cycle, at which HCQ is likely to exert its antiviral activity, time-of-drug-addition assays, such as SPR and MTT, were elaborated.

The experiment involved Vero E6 cells (3 × 10^4^ cells/mL) infected with HCoV-229E (MOI = 0.1) and treated with an HCQ standard (Sigma-Aldrich) at a concentration of 1 mg/mL (maximum non-toxic concentration). The replication cycle of HCoV-229E has demonstrated rapid viral propagation inside host cells, reaching maximum levels at 24 h *p.i.* [27]. This was confirmed by the AFM scans performed in our study—Figure 7, as well.

The inhibition of post-translational glycosylation with subsequent reduction in SARS-CoV-2 binding to and fusion with the angiotensin-converting enzyme 2 (ACE2) receptor of the host cell is an important antiviral effect of HCQ used in the treatment of SARS-CoV-2 infections [28]. Cleavage of SARS-CoV-2 spike (S) proteins by HCQ in the autophagosomes has also been reported [29]. Thus, the highest antiviral activity has to be expected at the stage of virus attachment to cells.

Indeed, our MTT assay confirmed the same mechanism of action in HCoV-229E: HCQ inhibited its replication in Vero E6 cells until 18 h *p.i.* A maximum antiviral activity was observed around 18 h—Figure 8. This coincided with the local maximum of virus control established by SPR (Figure 5), which confirmed that HCQ effectively inhibited virus replication at the stage of its attachment to the membrane and penetration into the cells.

The SPR study showed that maximum antiviral activity was reached around 12 h. This is also the stage of the virus replication cycle corresponding to the attachment to the membrane and penetration into the cells. The temporal shift against the MMT results could be explained by the method of compensation of SPR signals generated by the processes of cell growth and virus growth, as expanded in Section 2. The compensation procedure accumulates the error of surface plasmon resonance measurements (about 1.5 nm) and influences the peak position. However, the SPR signal is well pronounced at the expense of decreased accuracy.

## 4. Conclusions

A cell-based SPR assay was used to study cell growth, virus growth kinetics, and hydroxychloroquine antiviral kinetics. The MTT method was used as a reference since it is widely adopted for assessing cell metabolic activity. The kinetics revealed by the cell-based SPR assay was consistent with the findings of the MTT assay. Although the principles of the SPR and MTT methods are very different, the results obtained were very similar. All that showed that cell-based SPR is a reliable tool in investigating virus–host cell kinetics and antiviral drug activity. As expected, we found that the SPR assay provides better time resolution than MTT.

To the best of our knowledge, the present study is the first one focusing on the inhibiting effect of HCQ on the HCoV-229E virus. Both the SPR and MTT assay revealed that the antiviral efficiency is highest at the first stages of infection. 

However, a few points have to be considered for the correct SPR measurement. First, the cells have to be seeded on the SPR slides with almost uniform density, which would ensure a reliable SPR signal across the slide. Additionally, the cell density has to prevent light scattering so that the reflection from the grating can be detected reliably. In addition, the method for compensating the signals generated by the processes of cell and virus growth has to be carefully considered.

There is a significant limitation to the cell-based SPR assay in investigating virus–host kinetics—it cannot be applied for a period lasting more than two virus replication cycles. After this period, the SPR signal is masked by destruction of the cell monolayer, detachment from the grating surface, and presence of remnants of the destroyed cells on the surface. As a result, the SPR signal is not unambiguously defined by the virus–host interaction.

In the present research, we showed that the cell-based SPR assay is applicable for in vitro studies. However, an extension of the SPR assay for in vivo application is a matter of engineering solutions.

## Figures and Tables

**Figure 1 biosensors-11-00251-f001:**
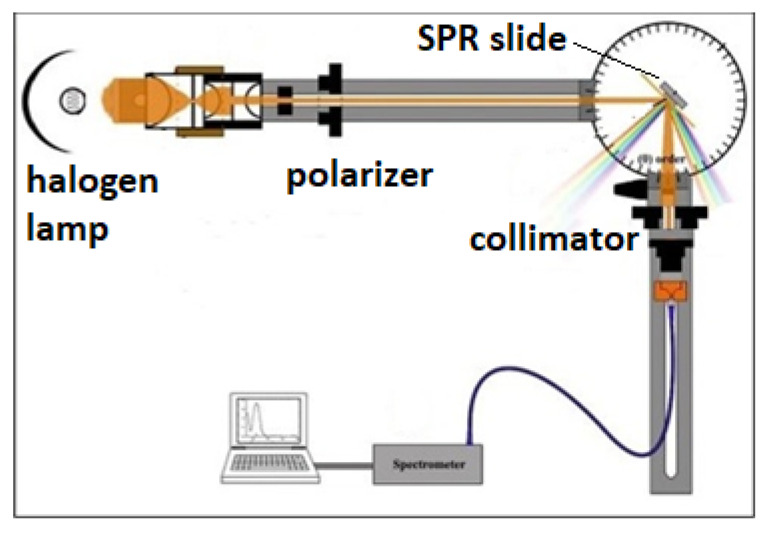
Optical setup for the SPR measurements.

**Figure 2 biosensors-11-00251-f002:**
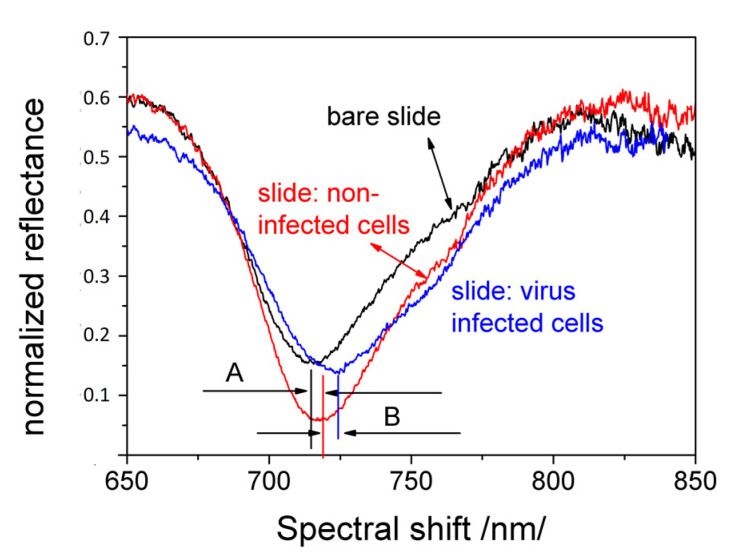
Experimentally observed resonances of bare grating and cell-based SPR assays.

**Figure 3 biosensors-11-00251-f003:**
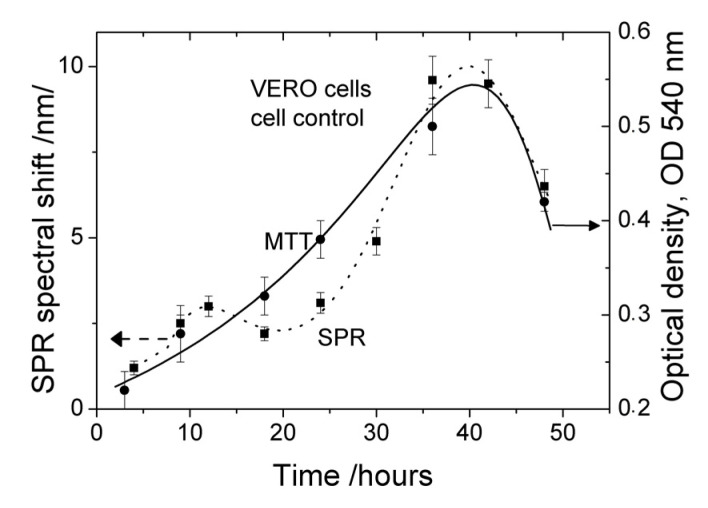
Growth kinetics of Vero cells: dotted line—SPR results (cell control); solid line—MTT results.

**Figure 4 biosensors-11-00251-f004:**
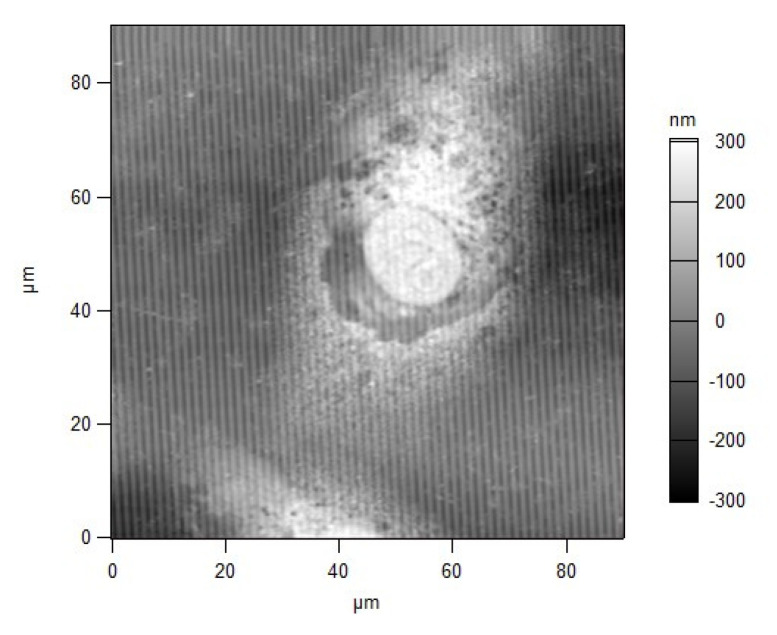
Vero cell at 24 h *p.i*.; AFM scan of the diffraction grating.

**Figure 5 biosensors-11-00251-f005:**
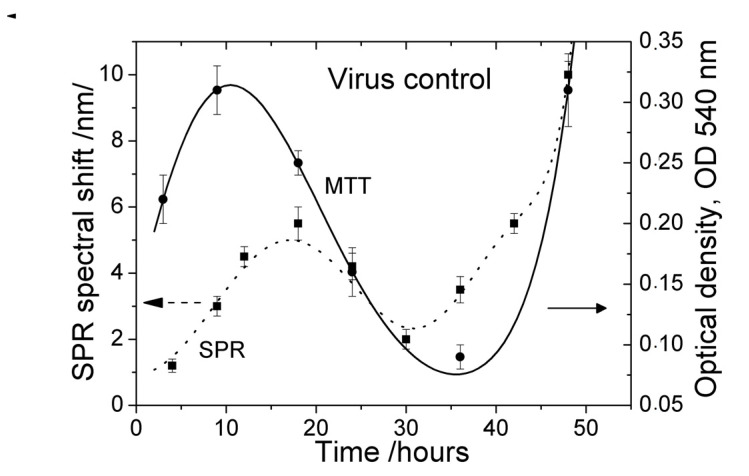
Growth kinetics of HCoV-229E-infected cells: dotted line—SPR results (virus control); solid line—MTT results.

**Figure 6 biosensors-11-00251-f006:**
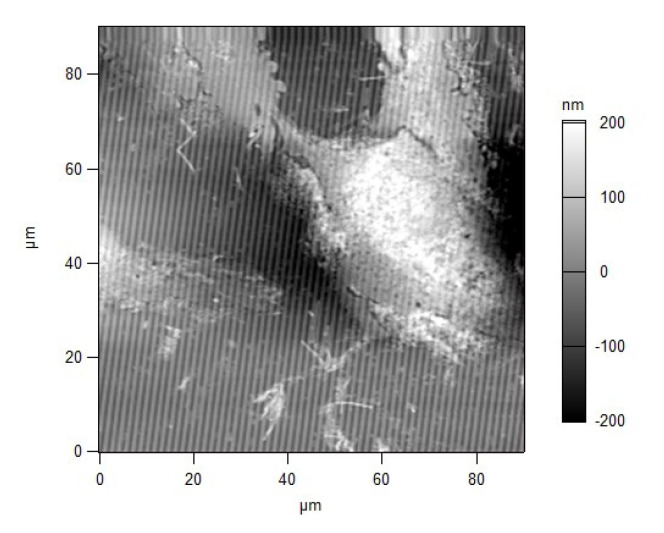
Infected Vero cell at 42 h *p.i*.; AFM scan of the diffraction grating.

**Figure 7 biosensors-11-00251-f007:**
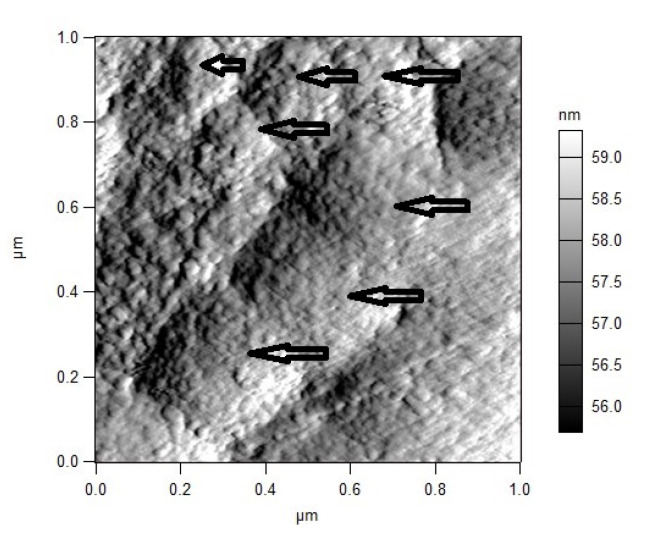
Viruses expressed in the intercellular space at 24 h *p.i*.

**Figure 8 biosensors-11-00251-f008:**
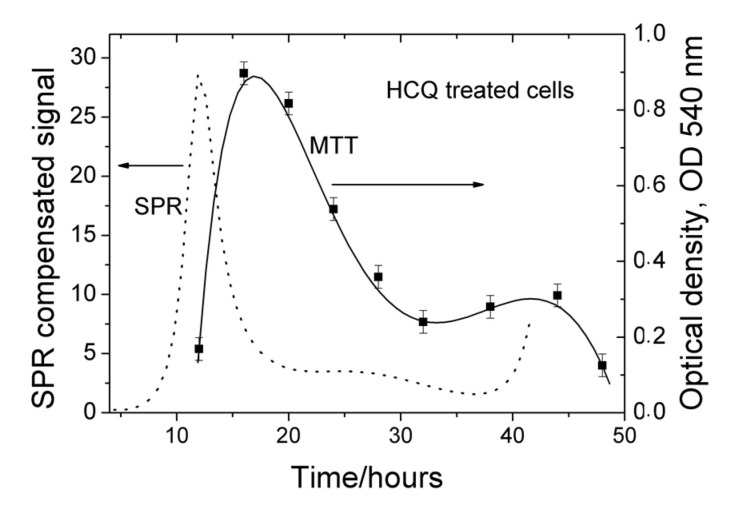
Kinetics of HCQ activity: dotted line—SPR results; solid line—MTT results.

## Data Availability

Data is contained within the article.

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
