# Peer review of "SPR-Based Kinetic Analysis of the Early Stages of Infection in Cells Infected with Human Coronavirus and Treated with Hydroxychloroquine"

_biosensors, 2021, doi:10.3390/bios11080251_

Round 1
Reviewer 1 Report
This article reports a cell-based SPR assay to be applied in the kinetic analysis of the early stages of viral infection of cells and the drug antiviral activity in the infected cells. The study is interesting and publishable.
Author Response
I would like to thank the reviewer for his favorable review.
Reviewer 2 Report
Report on the Manuscript Number: biosensors-1286860
Title: SPR-based kinetic analysis of the early stages of infection in cells infected with human coronavirus and treated with hydroxychloroquine
by Petia Genova-Kalou et alii
The authors describe a new SPR cell-based assay for examination of virus-host cell interactions and drug discovery processes. This approach allows for a more physiological setting compared to biochemical assays and was used to study the viral growth kinetics and to predict drug antiviral response in cells. The biological system is quite interesting in this pandemic period: cells infected with a human corona virus HCov-229E and treated with hydroxychloroquine. After a careful evaluation of the manuscript, I think that the paper needs major revisions to be considered for a publication on Biosensors MDPI for the following reasons:
1) I would comment extensively on the SPR working in the UV wavelength region. There is no mention of the operating wavelength of the optical sensor. Please provide more information and technical details about the optical set-up used in the experiments.
2) Due to the concerns exposed in Section 1.1 about penetration and detection depth, I would spend a whole Section to the technique used to bioconjugate the cells onto sensing surface. Please clarify and detail this crucial point.
3) Connected to the previous point: Could the authors show some numerical simulations of the structure grating+gold to enrich and complete the first part of the paper? In this way, it would be easier to comment the experimental data depicted in Fig. 1, 3 and 5.
4) Please define (for a better understanding) the naming of the MTT technique that is already present in the Abstract. The audience that is not in this field could be disoriented by such acronim.
5) Figure 6 is not at all clear. The description in the main text does not match with the AFM image. I would suggest to better treat this crucial point in the work.
6) The language quality should also be improved.
Author Response
- I would comment extensively on the SPR working in the UV wavelength region. There is no mention of the operating wavelength of the optical sensor. Please provide more information and technical details about the optical set-up used in the experiments.
Our experiment was carried out in the visible range 680-780 nm. Bearing in mind the well known relations of penetration depth [H. Reather, Surface plasmons on smooth and rough surfaces and gratings”, Springer-Verlag, 1988] and our knowledge of the refractive index of tissue [R. Khan, B. Gul, Sh. Khan, H. Nisar, I. Ahmad, Refractive index of biological tissues: Review, measurement techniques, and applications, Photodiagnosis and Photodynamic Therapy, Volume 33, March 2021, 102192] one can conclude that the penetration depth (of a plasmon wave excited in gold) is in the range 300-500 nm. The aim of our paper is to show:
- that despite such a shallow penetration depth (compared to cell height) SPR can provide information about biological events in the whole cell – a result in line with those reported in [13-15];
- Cell-based SPR assays can be applied successfully in studying virus-cell interactions.
We do not comment the pros and cons of UV-based SPR.
We fully agree with the reviewer that reference [24] related to the description of our optical set-up does not provide enough information.
We include new paragraphs and two new figures, line 161-183, with detailed description of the optical set-up and the SPR registration.
2) Due to the concerns exposed in Section 1.1 about penetration and detection depth, I would spend a whole Section to the technique used to bioconjugate the cells onto sensing surface. Please clarify and detail this crucial point.
Our experience shows that the grating surface has a crucial role for the effectiveness of the cell-based SPR assay since it increases the surface and ensures good adhesion. We comment on this point on line 208-209.
We provide additional information regarding the way in which cells were seeded and cultivated on the SPR slide on lines 140-160.
3) Connected to the previous point: Could the authors show some numerical simulations of the structure grating+gold to enrich and complete the first part of the paper? In this way, it would be easier to comment the experimental data depicted in Fig. 1, 3 and 5.
Unfortunately, numerical simulations are of no great avail. Even sophisticated software treating multilayer grating structure cannot provide reliable information in the case of spectral readout. The problem originates in the reflection of light from the interface metal/polycarbonate. The angular readout is not influenced by the reflection. However, the spectral readout is strongly influenced due to the dispersion characteristics of the grating and polycarbonate. We would like to mention that we are the only team to use spectral readout in grating-based SPR, to the best of our knowledge. We have studied in detail the problem and the results will be published in September. The reflection depends crucially on the technological parameters of the gold layer deposition, for example - the temperature of the polycarbonate substrate during the deposition. Also, an additional sublayer that absorbs or reflects effectively the light is highly desirable.
Briefly, the theoretical models cannot provide information on the characteristics of the plasmon wave and cannot help in interpreting the experimental results. For this reason, we interpreted the SPR response on the basis of MTT data (used as reference) and on the well-known kinetics of the HCoV-229E virus.
4) Please define (for a better understanding) the naming of the MTT technique that is already present in the Abstract. The audience that is not in this field could be disoriented by such acronim.
We provide the required information in Section 2: lines 129-134
5) Figure 6 is not at all clear. The description in the main text does not match with the AFM image. I would suggest to better treat this crucial point in the work.
We agree to the reviewer’s opinion – the reference to the figure does not match with the context. Now we refer to the figure on line 247. The virus expression is experimentally observable and is an essential point in its kinetics. With this figure we confirm the correctness of our interpretation of the SPR results.
Reviewer 3 Report
This article proposes the use of SPR as a tool to investigate cell growth/death kinetics without the need for complex labelling. The article needs major changes before being published that are summarized as follow:
- A major issue in this article is the focus on HCQ (invitro activity against COVID-19), at this stage this feels pointless with the controversy over the use of HCQ for COVID-19. It would be of a great benefit it to the paper if a different model is used. There are hundreds of publications that would contradict the findings in this article as the mode of action of the HCQ on Covid-19 is not yet well-established or agreed on.
- The manuscript did not provide any discussion on impedance-based cell assays which are also label free. A comparison between the optical and electrochemical methods should be included and authors should explain what advantage an optical method like SPR would provide over a similar electrochemical assay.
- Manuscript used SPR spectral shift as a measure to quantify virus kinetics; It is not clear what the spectral shift represents here and in what units, Is this a shift in critical angle or intensity? A clearer description of the measurements needs to be included in the manuscript (may be SI file; How the Au CD-R disk was prepared and treated? How it is fitted into the SPR? A schematic would be of a great value here.
- Why AFM was used to characterize the cells; does not bright field microscope provide better understanding of the cell phase and shape?
- Line 59 need recasting.
- Line 71-73 needs references
- Line 98 representative examples of SPR to study intracellular processes should be demonstrated.
Author Response
- A major issue in this article is the focus on HCQ (invitro activity against COVID-19), at this stage this feels pointless with the controversy over the use of HCQ for COVID-19. It would be of a great benefit it to the paper if a different model is used. There are hundreds of publications that would contradict the findings in this article as the mode of action of the HCQ on Covid-19 is not yet well-established or agreed on.
This remark is not adequate. We studied antiviral activity of HCQ against HCoV-229E virus – not SARS-CoV- 2, as we point out on lines: 20, 111, 114, 119, 127, 225, 272, 317.
- The manuscript did not provide any discussion on impedance-based cell assays which are also label free. A comparison between the optical and electrochemical methods should be included and authors should explain what advantage an optical method like SPR would provide over a similar electrochemical assay.
Yes, we do not provide information regarding electrochemical assay since the purposes of the paper are different, namely:
- To show that despite the shallow penetration depth of the plasmon wave (compared to cell height) SPR can provide information about biological events in the whole cell – a result in line with those reported in [2-4];
- To show that cell-based SPR assays can be successfully applied in studying virus-cell interactions.
There are many review papers and books (for example: Ye Fang, Label-Free Biosensor Methods in Drug Discovery, Springer, 2015, ISSN 1557-2153) which discuss comprehensively the pros and cons of label free methods. We would hardly be able to add more information to what is already available.
- Manuscript used SPR spectral shift as a measure to quantify virus kinetics; It is not clear what the spectral shift represents here and in what units, Is this a shift in critical angle or intensity? A clearer description of the measurements needs to be included in the manuscript (may be SI file; How the Au CD-R disk was prepared and treated? How it is fitted into the SPR? A schematic would be of a great value here
We fully agree with the reviewer that reference [24] related to the description of our optical set-up does not provide enough information.
Now we have included new paragraphs and two new figures, lines 161-183, with a detailed description of the optical set-up and the SPR registration.
- Why AFM was used to characterize the cells; does not bright field microscope provide better understanding of the cell phase and shape?
Yes, an optical microscopic study provides enough information on the cell morphology changes. We used AFM for two reasons:
- To show that cells follow the grating surface. The tight cell attachment to the slide surface provides an effective plasmon penetration into the cells increasing RI changes detectable by SPR. Lines 208-209 address this remark of the reviewer.
- HCoV-229E virus is 150-200 nm in size (shown in Fig. 7) and cannot be observed by a conventional optical microscope. Actually, a picture in Fig.7 is a magnified part of a picture in Fig.6 and shows the expression of viruses near the cell membrane.
- Line 59 need recasting
We have clarified the sentence: lines 59-61
- Line 71-73 needs references
Corrected: new references 16-18 are included
- Line 98 representative examples of SPR to study intracellular processes should be demonstrated.
We do not agree with this reviewer’s requirement. Section 1 summaries the main results achieved in cell-based SPR sensing of intracellular processes.
Line 55 refers to [2-4] which address sensing of intracellular biological events in the area near the plasma membrane.
Lines 57,58 refer to [6-8] which address sensing of intracellular biological events in the upper area of cells.
Lines 64-66 refer to [8-15] reporting results that despite the shallow penetration depth (compared to cell height) SPR can provide correct information about biological events in the whole cell.

Round 2
Reviewer 2 Report
No further comments.
Reviewer 3 Report
I think the updated versionof the paper is now suitable for publication in biosensors.